# Detection of Canine Urothelial Carcinoma Cells in Urine Using 5-Aminolevulinic Acid

**DOI:** 10.3390/ani12040485

**Published:** 2022-02-16

**Authors:** Kenjiro Kaji, Tomohiro Yonezawa, Yasuyuki Momoi, Shingo Maeda

**Affiliations:** Department of Veterinary Clinical Pathobiology, Graduate School of Agricultural and Life Sciences, The University of Tokyo, Tokyo 113-8657, Japan; kajiv10037@g.ecc.u-tokyo.ac.jp (K.K.); ayone@g.ecc.u-tokyo.ac.jp (T.Y.); momoi@g.ecc.u-tokyo.ac.jp (Y.M.)

**Keywords:** dog, transitional cell carcinoma, 5-aminolevulinic acid

## Abstract

**Simple Summary:**

5-aminolevulinic acid (5-ALA) is a natural amino acid that is metabolized in the mitochondria and used in the synthesis of heme. In human medicine, the fluorescence property of 5-ALA has been used for photodynamic tumor diagnosis. Nevertheless, research on 5-ALA in veterinary medicine is very limited. In this study, we aimed to evaluate the efficacy of 5-ALA to detect canine urothelial carcinoma and to clarify its diagnostic accuracy. The addition of 5-ALA to tumor cells from the urine of patients with urothelial carcinoma and canine urothelial carcinoma cell lines caused red fluorescence, and the amount of fluorescence was significantly higher than that in healthy dogs. Additionally, cases with high fluorescence intensity had more tumor invasion and metastasis. This study showed that 5-ALA can be used to detect canine urothelial carcinoma cells in urine with relatively high diagnostic accuracy.

**Abstract:**

This study aimed to establish a method to detect canine urothelial carcinoma cells in urine using 5-aminolevulinic acid (5-ALA) and to evaluate its diagnostic accuracy. Urine samples were collected from 21 dogs diagnosed with urothelial carcinoma and three urothelial carcinoma cell lines were used. Urine samples obtained from seven healthy dogs were used as controls. Cells in the urine sediment, or urothelial carcinoma cell lines, were cultured with 5-ALA and then observed under a fluorescence microscope. Moreover, we examined the relationship between fluorescence intensity and the presence of metastasis as well as tumor invasion into the bladder wall in cases of urothelial carcinoma. Urine-derived cells from urothelial carcinoma and urothelial carcinoma cell lines showed clearer red fluorescence with the addition of 5-ALA compared to that exhibited by the cells from healthy dogs. The sensitivity and specificity of the diagnosis of urothelial carcinoma were 90% and 86%, respectively. Significant associations were found between fluorescence intensity and tumor metastasis and bladder wall invasion. This study showed that 5-ALA can be used to detect urothelial carcinoma cells in dogs with relatively high diagnostic accuracy. Further, the fluorescence intensity of tumor cells caused by 5-ALA correlated with the clinical condition of urothelial carcinoma cases, which suggested that 5-ALA could be used as a prognostic marker for canine urothelial carcinoma.

## 1. Introduction

Canine urothelial carcinoma, also known as transitional cell carcinoma, is a malignant tumor characterized by local invasion and metastasis to the lymph nodes and distant sites [1]. Diagnosis of canine urothelial carcinoma is often delayed due to nonspecific initial symptoms, and it has been reported that approximately 50% of cases have distant metastases at autopsy [2]. Therefore, there is a need for effective medical treatment of urothelial carcinoma, as well as simple and accurate diagnostic methods for early detection of these tumors. The most reliable test to diagnose urothelial carcinoma is histopathology, but it is a highly invasive test. The detection of abnormal epithelial cells in urine collected from patients by cytology can also aid in diagnosis [2,3]. However, cytology alone cannot diagnose canine urothelial carcinoma because normal urothelial cells can also become morphologically abnormal, similar to tumor cells, due to prolonged contact with urine or inflammation [4]. The detection of BRAF mutations in urine-derived cells is also widely used as a method to support the diagnosis of urothelial carcinoma in dogs. BRAF is a type of RAF protein belonging to the RAF/MEK/ERK mitogen-activated protein kinase (MAPK) pathway that contributes significantly to cell survival and proliferation [5]. Previous studies have described a somatic point mutation in the BRAF gene (BRAF^V595E^) in canine urothelial carcinoma cells, and detection of this mutation can be used as a diagnostic aid [1,6,7]. However, as only approximately 60–70% of urothelial carcinoma cases carry the BRAF^V595E^ mutation [7], this test alone is considered insufficient for detecting urothelial carcinoma. Consequently, a reliable, non-invasive diagnostic method is necessary for the early detection of urothelial carcinoma in dogs.

5-aminolevulinic acid (5-ALA) is a natural amino acid that can either be biosynthesized in the body or ingested [8]. In the body, 5-ALA is synthesized in the mitochondria from glycine and succinyl-CoA via aminolevulinic acid synthase. Synthesized 5-ALA is metabolized to porphobilinogen and finally to protoporphyrin IX (PpIX) [9]. PpIX biosynthesized in normal cells coordinates with iron ions to form heme, a component of hemoglobin in the blood, and P450, a drug-metabolizing enzyme [9]. Conversely, in tumor cells such as gliomas [10], digestive tract cancers [9,11], and bladder cancers [12], insufficient iron oxidation in the mitochondria hampers heme synthesis, which leads to the accumulation of the heme precursor PpIX.

PpIX emits red fluorescence in the 660 to 720 nm range when excited by blue–violet light at wavelengths from 365 to 430 nm. In human medicine, using this property of 5-ALA, photodynamic diagnosis is widely performed by administering 5-ALA to patients with bladder cancer [13] or gastric cancer [14] and irradiating them with blue–violet light to detect red fluorescence in the malignant lesion. 5-ALA has also been used to detect urine-derived tumor cells in urogenital cancers, such as bladder [15] and prostate cancer [16]. Furthermore, PpIX produces reactive oxygen species (ROS) when excited by red light at approximately 635 nm [17] and can be used as photodynamic therapy for the treatment of gliomas [18] and gastric cancer [19].

Research on photodynamic diagnosis and treatment using 5-ALA in veterinary medicine is limited. A few studies have investigated the efficacy of PDT using 5-ALA in canine adenocarcinoma, mammary carcinoma, and urothelial carcinoma, with a combined sensitivity and specificity of 89.5% and 50% for all tumors, respectively [20]. However, the diagnostic accuracy for each tumor is unknown due to the small number of individual tumor cases. Therefore, photodynamic therapy with 5-ALA has not been used in a clinical setting at present. Considering this, the purpose of this study was to establish a method to detect canine urothelial carcinoma in urine using 5-ALA and to evaluate its diagnostic accuracy and potential clinical application. We also investigated the efficacy of 5-ALA as a prognostic marker by demonstrating the relationship between 5-ALA and pathological conditions such as tumor metastasis and invasion of the muscle layer.

## 2. Materials and Methods

### 2.1. Study Population and Samples

Urine samples were collected from 21 dogs diagnosed with urothelial carcinoma by ultrasonography and histopathological examination at the University of Tokyo Veterinary Medical Center from April 2020 to April 2021. A total of 4 of the 21 patients had vesical trigone, 7 of 21 had urethra involvement, and 10 of 21 had prostate involvement. In these cases, there were no findings that would led us to suspect a urinary tract infection. The information for all 21 cases is presented in Table 1. Written informed consent was obtained from all the dog owners prior to sample collection. The dogs ranged in age from 10 to 15 years (median, 13 years). None of the dogs received chemotherapy or radiation therapy during the first visit. Ultrasonography and medical records were used to review the presence or absence of lymph node metastasis, distant metastasis, and muscle invasion of the tumor in each case. The tumor location was also assessed based on ultrasonography. BRAF^V595E^ mutations were examined in all cases using a digital PCR assay, as previously described [5]. Urine samples obtained from seven healthy dogs were used as controls. The healthy dogs were all 8-year-old neutered males and were confirmed to be free of disease by blood tests, urinalysis, and ultrasonography. Urine samples were collected from cases and healthy dogs using a 4-6Fr catheter (Atom Medical Corporation, Tokyo, Japan) under ultrasound guidance. The urine sample was centrifuged at 1500× *g* for 5 min to separate the urine sediment. Subsequently, red blood cells (RBCs) were removed using RBC Lysis Buffer (PluriSelect Life Science, Leipzig, Germany). All procedures in this study were conducted in accordance with the guidelines of the Animal Care Committee of the University of Tokyo (approval number: P17-108).

### 2.2. Cell Lines

Three urothelial carcinoma cell lines (Love, Sora, TCCUB) [21] were used for the experiments. The cell lines were grown in RPMI medium (Wako, Osaka, Japan) containing 10% fetal bovine serum (FBS; Gibco, Carlsbad, CA, USA) and 1% penicillin—streptomycin (Gibco)—at 37 °C with 5% CO_2_.

### 2.3. Cell Culture with 5-ALA

The urine sediment, from which RBCs were removed, was washed with phosphate-buffered saline (PBS) and centrifuged three times at 1500 rpm for 3 min. The number of cells in the urine sediment was adjusted to 5 × 10^3^. Based on previous reports [15], cells were cultured in RPMI medium without FBS and 1 mM 5-ALA (Neopharma Japan, Tokyo) or vehicle (ultra-pure water) for 2 h at 37 °C and 5% CO_2_. All urine sediment samples were incubated with 5-ALA within six hours of harvesting. The three canine urothelial carcinoma cell lines were cultured with 5-ALA under the same conditions. The cultured urine sediment and cell lines were washed three times with PBS prior to the experiment.

### 2.4. Fluorescence Observation by 5-ALA

Urine sediment and cell lines cultured with 5-ALA or vehicle were observed under a fluorescence microscope (ECLIPSE E800, Nikon, Tokyo, Japan) at an excitation wavelength of 430 nm and a emission wavelength of 660 nm.

### 2.5. Quantification of Fluorescence by 5-ALA

Fluorescence was measured in urine sediment and cell lines cultured with 5-ALA or vehicle using a spectrofluorometer (Promega KK, Tokyo, Japan) at an excitation wavelength of 365 nm and an emission wavelength of 660 to 720 nm. The fluorescence intensity was evaluated using the relative ratio between the sample with the vehicle and that with 5-ALA.

### 2.6. Statistical Analyses

The unpaired Student’s t-test was used to examine the presence of the BRAF^V595E^ mutation, sex, and the fluorescence intensity of 5-ALA. Bonferroni’s multiple comparison test was used to compare the fluorescence intensity of ALA among healthy dogs, urothelial carcinoma cases, urothelial carcinoma cell lines, and among the no treatment, NSAIDs, and prednisolone groups. The accuracy of the detection of canine urothelial carcinoma cells in urine using 5-ALA was evaluated by calculating the area under the ROC curve (AOC). The diagnostic sensitivity and specificity using 5-ALA were evaluated by setting the cutoff value of the red fluorescence ratio to distinguish tumor cells from non-tumor cells, based on the ROC curve analysis. Comparisons of the fluorescence intensity using 5-ALA with tumor metastasis were performed using Fisher’s exact test. Statistical analyses were performed using GraphPad Prism, version 5.0.1 (GraphPad Software, San Diego, CA, USA).

## 3. Results

### 3.1. Canine Urothelial Carcinoma Cells Exhibited Red Fluorescence upon Addition of 5-ALA

Representative fluorescence microscopy images of urine-derived cells from healthy dogs, urothelial carcinoma cases, and urothelial carcinoma cell lines after the addition of 5-ALA or vehicle are shown in Figure 1. No red fluorescence was observed in any of the cells to which the vehicle was added. In urine-derived cells from healthy dogs, the addition of 5-ALA caused almost no red fluorescence. On the other hand, urine-derived cells from urothelial carcinoma cases had various degrees of red fluorescence in the cytoplasm when 5-ALA was added. Furthermore, upon addition of 5-ALA, distinct red fluorescence was observed in the cytoplasm of three canine urothelial carcinoma cell lines: Sora, Love, and TCCUB. In addition, when the red fluorescence observed in these cases was quantified using a plate reader, the fluorescence ratio was significantly increased in urothelial carcinoma cases (mean ± SD, 1.95 ± 0.15) and urothelial carcinoma cell lines (mean ± SD, 2.41 ± 0.11), compared to healthy dogs (mean ± SD, 1.24 ± 0.03) (Figure 2). There was no significant difference between urothelial carcinoma cases and urothelial carcinoma cell lines. Sex and treatment status (no treatment, NSAIDs, prednisolone) did not affect the fluorescence intensity of 5-ALA (Appendix A, Figure A1 and Figure A2). In addition, there was no significant correlation between tumor location (vesical torigone, urethra involvement, prostate involvement) and fluorescence intensity by 5-ALA (Appendix A, Figure A3).

### 3.2. 5-ALA-Induced Fluorescence Is Useful for the Diagnosis of Canine Urothelial Carcinoma

A ROC curve was generated based on the relative fluorescence ratios measured in urine-derived cells from urothelial carcinoma cases and healthy dogs, and the AUC was 0.87 (95% CI, 0.73–0.97) (Figure 3a). Based on this ROC analysis, a cutoff value of 1.29 was set (Figure 3b), and the diagnostic sensitivity and specificity of 5-ALA to detect canine urothelial carcinoma cells in urine were 90% and 86%, respectively, indicating high accuracy.

### 3.3. 5-ALA-Induced Fluorescence Intensity Was Significantly Associated with the Clinical Condition of Urothelial Carcinoma Cases

Canine urothelial carcinoma cases were divided into high and low fluorescence groups (Table 1) based on the median fluorescence ratios (median, 1.65) when 5-ALA was used. We were also examining whether ALA fluorescence correlated with the patient’s clinical condition, such as metastasis or invasion of the muscle layer. A total of 7 of the 21 patients had metastasis in the lymph nodes and distant organs. Invasion of the bladder wall was observed in 9 of the 21 cases. The high fluorescence group showed significantly more tumor invasion into the muscle layer and metastasis than the low fluorescence group (Table 2).

### 3.4. 5-ALA-Induced Fluorescence Intensity of Urine-Derived Cells Was Significantly Associated with the BRAF^V595E^ Gene Mutation in Urothelial Carcinoma Cases

The digital PCR data showed that 13 of the 21 cases had the BRAF^V595F^ mutations and 8 had the wild-type BRAF genes. The intensity of 5-ALA fluorescence was significantly higher in cases with the BRAF^V595E^ mutation than in those with wild-type BRAF (Figure 4).

## 4. Discussion

In this study, we demonstrated that urothelial carcinoma cells in canine urine samples could be detected using 5-ALA with relatively high diagnostic accuracy. In addition, our data suggest that the fluorescence intensity of tumor cells caused by 5-ALA was related to muscle invasion and tumor metastasis.

In human medicine, urogenital tumor cells in urine were detected using 5-ALA, with a sensitivity of 87% and specificity of 86% for bladder cancer, whereas the values were 74% and 70%, respectively, for prostate cancer [15]. The accuracy of photodynamic diagnosis using 5-ALA for various feline and canine tumor tissues, such as mammary and bladder cancer, was 89.5% sensitivity and 50% specificity [20]. Nonetheless, this was the first study evaluating tumor cell detection in urine using 5-ALA in dogs. In this study, canine urothelial carcinoma cells in urine were detected using 5-ALA with relatively high accuracy, a sensitivity of 90% and a specificity of 86%. These results highlighted the potential of 5-ALA in the establishment of a diagnostic system to detect canine urothelial carcinoma using urine samples.

Various studies have been conducted to understand why PpIX accumulates excessively in tumor cells compared to normal cells. Exogenously administered ALA is taken up by cells via active transporters such as peptide transporters 1,2 and GABA transporter 2 on the cell membrane [22,23,24]. 5-ALA is metabolized to PpIX by heme synthase [25]. PpIX is then synthesized into heme by ferrochelatase by binding to iron ions. In normal cells, exogenously administered ALA is rapidly metabolized, but in tumor cells iron ions are not oxidized efficiently and the precursor PpIX tends to accumulate in the cells [26,27,28]. This difference is attributed to the way tumor cells produce energy. Normal cells produce energy primarily through oxidative phosphorylation in the mitochondria, whereas various cancer cells produce energy through aerobic glycolysis, consuming large amounts of glucose and producing excess lactate, leading to cellular acidosis. As a result, tumor cells lack NADH and Fe^3+^ is not sufficiently converted to Fe^2+^, resulting in the accumulation of PpIX [29]. The accumulation of PpIX in malignant lymphoma cell lines may be due to increased activity of porphobilinogen deaminase, an enzyme that metabolizes 5-ALA, and decreased activity of ferrochelatase, an enzyme that binds PpIX to iron ions [30]. In addition, porphyrin compounds such as PpIX have a high affinity for lipoproteins, especially LDL proteins. This suggests that PpIX may be involved in the intracellular uptake of tumor cells with increased LDL receptor activity [31,32]. In this study, we found that the fluorescence intensity of 5-ALA in canine urothelial carcinoma cells correlated with tumor metastasis and invasion of the muscle layer. This suggested that the mechanisms involved in 5-ALA and PpIX metabolism may contribute to tumor pathogenesis. Nonetheless, the way these mechanisms contribute is unknown at this stage and will be clarified in future studies.

In this study, the fluorescence intensity of 5-ALA was significantly higher in cases with the BRAF^V595E^ mutation. There are no studies on the direct relationship between BRAF mutations and 5-ALA, although a few studies describe the relationship between MEK, a downstream signal of BRAF, and the fluorescence of 5-ALA [33,34]. In a previous study, intracellular accumulation of PpIX increased in RasV12-transformed NIH3T3 cells after addition of an MEK inhibitor. Additionally, the fluorescence intensity was enhanced when 5-ALA was added. This was because MEK contributes to increased expression of ABCB1, a drug efflux transporter that releases PpIX from tumor cells, and to the activity of ferrochelatase, which converts PpIX to heme [33,34]. Increased ABCB1 expression and enhanced PpIX to heme conversion may reduce PpIX accumulation since the signal to MEK is enhanced in cases with BRAF mutations. However, the present study showed that the fluorescence intensity of 5-ALA was higher in cases with BRAF mutations. The reason for this phenomenon remains unclear. Possible explanations include differences in animal species. Moreover, the downstream pathway between signals initiated by Ras mutation and those initiated by BRAF^V595E^ mutation may also be different.

This study has some limitations. First, the sample size was relatively small. Second, in this study, the presence or absence of tumor invasion in the muscle layer was evaluated only by ultrasonography, which may have missed invasion that was not actually visible on the images. Since pathological examination of all layers of the bladder is necessary to truly evaluate invasion into the muscular layer, it is necessary to increase the number of cases and evaluate them in combination with detailed pathological examination in the future. Third, we did not examine urological diseases other than urothelial carcinoma. In human medicine, bladder polyps, a benign neoplastic disease, and cystitis, an inflammatory disease, may cause false positives in fluorescence endoscopy using 5-ALA because they emit fluorescence similar to that of bladder cancer [35]. In veterinary medicine false positives can be caused by benign skin appendage tumors, plasmacytoma, and histiocytoma fluoresce with 5-ALA [20]. Therefore, canine cystitis and bladder polyps may cause false positives in urothelial carcinoma diagnosis using 5-ALA. This may have decreased the sensitivity and specificity of the test. Since this study was a pilot study, we plan to increase the number of cases to see if we can differentiate tumors from other urinary tract diseases.

## 5. Conclusions

This study showed that urine-derived tumor cells could be detected with high sensitivity and specificity using 5-ALA in canine urothelial carcinoma. In addition, it was found that fluorescence intensity of 5-ALA might correlate with disease progression, suggesting that 5-ALA-induced fluorescence is a prognostic marker for canine urothelial carcinoma.

## Figures and Tables

**Figure 1 animals-12-00485-f001:**
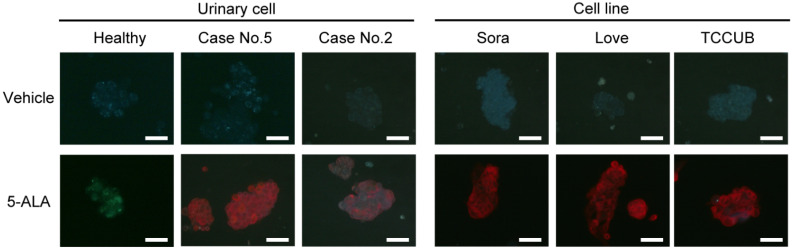
The top panel shows fluorescence microscopy images of urine-derived cells from healthy dogs, urothelial carcinoma cases (No. 2 and No. 5), and urothelial carcinoma cell lines (Sora, Love, TCCUB) when vehicles were added. No red fluorescence was observed in any of the cells. The lower panel shows the fluorescence microscopy image when 5-ALA was added. No red fluorescence is observed in urine-derived cells from healthy dogs. The urine-derived cells from case No. 5 showed clear red fluorescence. Weak red fluorescence was observed in urine-derived cells derived from case No. 2. Strong red fluorescence was observed in all three urothelial carcinoma cell lines: Sora, Love, and TCCUB. Bar = 50 μm.

**Figure 2 animals-12-00485-f002:**
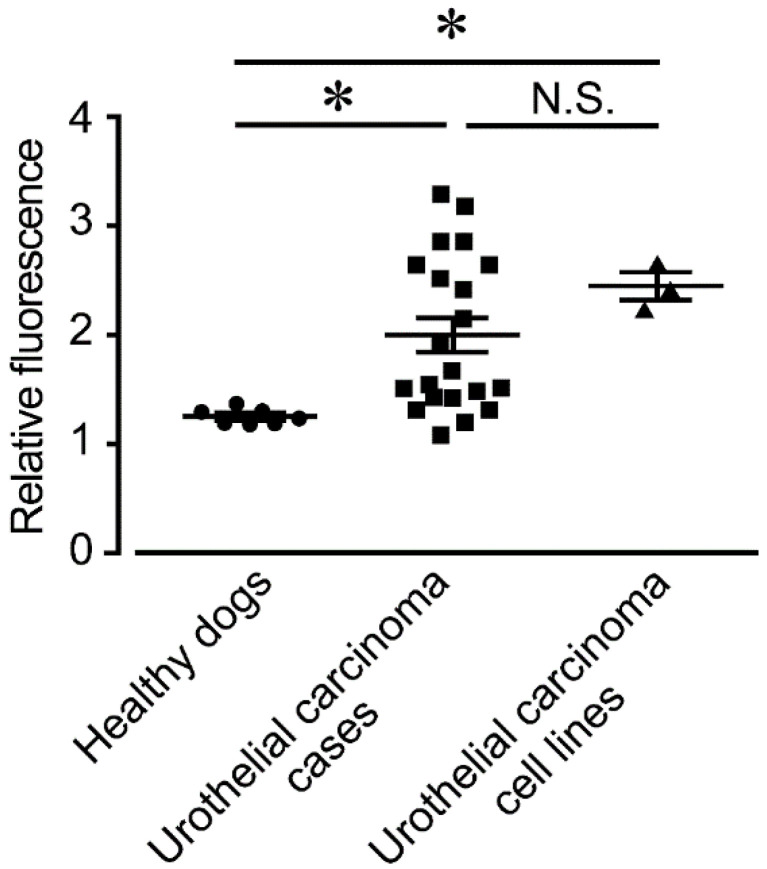
Fluorescence intensity in healthy dogs (*n* = 7), urothelial carcinoma cases (*n* = 21), and urothelial carcinoma cell lines (*n* = 3). Fluorescence intensity is expressed as a ratio of the amount of fluorescence in the 5-ALA supplemented group relative to the amount of fluorescence in the vehicle group for each sample. Statistical significance was determined by Bonferroni’s multiple comparison test. * *p* < 0.05. The horizonal lines indicate the mean ± SD.

**Figure 3 animals-12-00485-f003:**
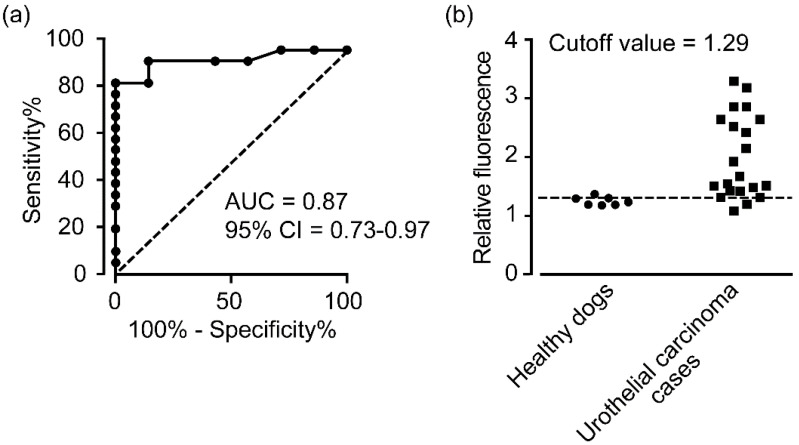
(**a**) ROC curves for the fluorescence intensity of 5-ALA in healthy dogs (*n* = 7) vs. urothelial carcinoma cases (*n* = 21). AUC—area under the curve. (**b**) The cutoff value was calculated by ROC curve analysis. Fluorescence intensity is expressed as a ratio of the fluorescence in the 5-ALA supplemented group relative to the fluorescence in the vehicle group for each sample. The horizontal dot line indicates the cutoff value.

**Figure 4 animals-12-00485-f004:**
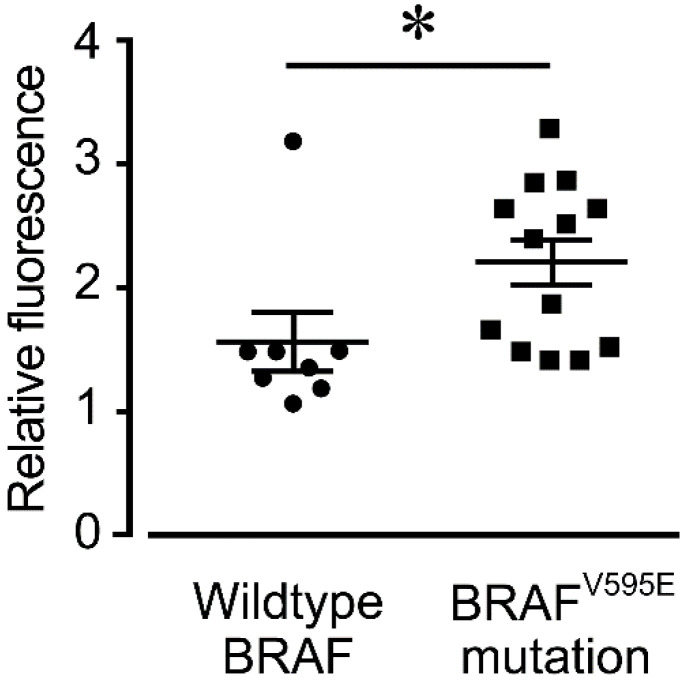
Fluorescence intensity in urothelial carcinoma cases with wild-type BRAF (*n* = 8) and cases with the BRAF^V595E^ mutation (*n* = 13). Fluorescence intensity is expressed as a ratio of the amount of fluorescence in the 5-ALA supplemented group relative to the amount of fluorescence in the vehicle group for each sample. * *p* < 0.05. The horizontal lines indicate the mean ± SD.

**Table 1 animals-12-00485-t001:** Profiles of the dogs with urothelial carcinoma in this study.

Case No.	Age (Years)	Sex	Breed	BRAF^V595E^ Mutation	5-ALA Fluorescence Intensity	Tumor Location	Metastasis	Muscle- Invasion	Medication Status at First Visit
1	14	FS	Chihuahua	+	High	Prostate involvement	+	+	Prednisolone
2	14	FS	Pug	+	Low	Urethra involvement	-	-	Piroxicam
3	14	MN	Toy Poodle	+	High	Prostate involvement	-	-	Piroxicam
4	13	FS	Border collie	-	Low	Vesical torigone	+	+	Piroxicam
5	11	FS	Miniature Schnauzer	-	High	Urethra involvement	+	+	No treatment
6	12	MN	Mix	+	High	Prostate involvement	+	+	Prednisolone
7	12	FS	Toy Poodle	+	High	Vesical torigone	-	+	No treatment
8	15	M	Miniature Pinscher	-	Low	Urethra involvement	-	-	No treatment
9	12	MN	Shetland Sheepdog	-	Low	Urethra involvement	-	-	Piroxicam
10	11	MN	Miniature Schnauzer	+	High	Prostate involvement	-	-	Prednisolone
11	12	MN	Cavalier King Charles Spaniel	+	High	Urethra involvement	+	+	No treatment
12	13	FS	Mix	+	High	Urethra involvement	-	-	No treatment
13	14	MN	Toy Poodle	+	High	Prostate involvement	+	+	Piroxicam
14	14	MN	Chihuahua	-	Low	Prostate involvement	-	-	No treatment
15	10	MN	Cairn Terrier	+	Low	Prostate involvement	-	+	Piroxicam
16	13	M	Toy Poodle	+	Low	Prostate involvement	-	-	Firocoxib
17	12	M	Mix	-	Low	Urethra involvement	-	-	Piroxicam
18	15	MN	Shih Tzu	+	Low	Prostate involvement	-	-	Piroxicam
19	12	MN	Papillon	+	High	Vesical torigone	+	+	No treatment
20	14	FS	Maltese	-	Low	Urethra involvement	-	-	Piroxicam
21	13	MN	Toy Poodle	-	Low	Prostate involvement	-	-	Firocoxib

Abbreviations: F, female; FS, spayed female; M, male; MN, neutured male.

**Table 2 animals-12-00485-t002:** Association between fluorescence intensity of 5-ala and clinical condition of urothelial carcinoma cases.

			**Fluorescence Intensity**		
		**Cases (%)**	**Low**	**High**	***p*-Value**
**Muscle-invesion**	Present	9 (45)	2	7	0.030
	Absent	12 (55)	9	3	
**Metastasis**	Present	7 (34)	1	6	0.019
	Absent	14 (66)	10	4	

Fisher’s exact test was used to analyze the above data.

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
