# Peer review of "Detection of Canine Urothelial Carcinoma Cells in Urine Using 5-Aminolevulinic Acid"

_animals, 2022, doi:10.3390/ani12040485_

Round 1

Reviewer 1 Report

Page 2, line 50-51 “Detection of BRAF mutations from urine-derived cells is also widely used to diagnose canine urothelial carcinoma.” This statement should be revised to state that the presence & detection of BRAF mutations support a diagnose of canine urothelial carcinoma.  A positive BRAF test is not equivalent to a definitive diagnosis of urothelial carcinoma, and reports of positive BRAF tests in dogs with no visible lesions within their urinary tract are increasing.  As the authors point out in just a few sentences, not all canine urothelial carcinomas have BRAF mutations, thus this test should not be relied upon as a sole diagnostic test.  I would argue that bladder ultrasound alone may be the most cost-effective and least invasive test that can be highly sensitive for detecting urothelial carcinoma.

Page 2, line 80 & 84 – “transitional cell carcinoma” – it may be important for clarity’s sake for the authors to include a parenthetical that transitional cell carcinoma and urothelial carcinoma are the same disease but terminology has changed since the PDT studies were published.

Page 3, line 106 – “bladder wall invasion” – Can ultrasound truly be used to determine if invasion into the bladder wall is present?  Is it possible that some of the lesions classified as not muscle invasive were in fact muscle invasive but lacked a large enough invasion to be termed that via ultrasound exam alone?

Table 1. It appears that ALA intensity was low in 8/11 dogs treated with an NSAID vs. 2/7 dogs receiving no treatment vs 0/3 in dogs receiving prednisolone.  Were dogs assessed for differences between groups?  For male dogs, the ALA intensity was low in 8/14 dogs and for female dogs, ALA intensity was low in 3/7 dogs.  Were there significant differences based on sex?

Page 7, line 230: “we demonstrated that urothelial carcinoma cells in canine and human urine samples can be detected…”  Were there samples from humans tested?

Figure 2. This data should be broken up by treatments the dogs with urothelial carcinoma received (NSAID, prednisolone or no treatment) to determine if the treatment influenced the results.  There is a much wider range in the clinical dogs with urothelial carcinoma vs. the urothelial carcinoma cell lines, and it may be important to establish if there was any influence of the treatments these dogs already received.  This is particularly important given the known anti-tumor activity of NSAIDs in urothelial carcinoma and the widespread use of these medications in veterinary medicine once a dog is diagnosed with a suspected bladder tumor.

Table 2. Again, I’d like clairification on how muscle-invasiveness was determined.  If no full thickness biopsy specimens were available, I’m concerned that US alone may underestimate muscle invasion.  It also may be important to evaluate dogs with prostatic involvement separately as this has historically been associated with a worse prognosis.

With respect to the discussion and conclusions, I’m concerned that not enough variables were evaluated to determine if the high/low intensity is related to BRAF status, treatment (which many dogs had already had), tumor location (e.g., prostatic involvement), etc.  As with many studies looking to use urine for the purposes of diagnosing urothelial carcinoma, I am concerned about the lack of a group with inflammation (cystitis) and/or urinary tract infection present.  I would suggest evaluating for the same thing in groups of dogs with both of these conditions (or at least urinary tract infections).  Did any of the dogs with urothelial carcinoma have active urinary tract infections?  I think these questions need to be addressed before the discussion can truly be reviewed.  The authors acknowledge this as a limitation on page 8 beginning a line 283 and understand that benign conditions may mimic urothelial carcinoma.  It is essential that these sorts of conditions be included when developing this sort of diagnostic aid, otherwise potential false positives will be missed and the utility of the test will be overestimated.  At a minimum, the authors should perform additional analysis in urine samples from dogs with urinary tract infections and report if any of the dogs in this study had urinary tract infections.  Urinary tract infections are common and it should not be difficult to collect urine from multiple dogs with this condition in a short time frame.

Author Response

Reviewer: 1

1) Page 2, line 50-51 “Detection of BRAF mutations from urine-derived cells is also widely used to diagnose canine urothelial carcinoma.” This statement should be revised to state that the presence & detection of BRAF mutations support a diagnose of canine urothelial carcinoma.  A positive BRAF test is not equivalent to a definitive diagnosis of urothelial carcinoma, and reports of positive BRAF tests in dogs with no visible lesions within their urinary tract are increasing.  As the authors point out in just a few sentences, not all canine urothelial carcinomas have BRAF mutations, thus this test should not be relied upon as a sole diagnostic test.  I would argue that bladder ultrasound alone may be the most cost-effective and least invasive test that can be highly sensitive for detecting urothelial carcinoma.   

[Response]

Thank you very much for your comments. The text of the relevant part was rewritten as follows.

“The detection of BRAF mutations in urine-derived cells is also widely used as a method to support the diagnosis of urothelial carcinoma in dogs.” P. 2, line 58-60.

2) Page 2, line 80 & 84 – “transitional cell carcinoma” – it may be important for clarity’s sake for the authors to include a parenthetical that transitional cell carcinoma and urothelial carcinoma are the same disease but terminology has changed since the PDT studies were published.  

[Response]

Thank you very much for your comments. Replaced "transitional cell carcinoma" with "urothelial carcinoma" in the text and added the following description.

“Canine urothelial carcinoma, also known as transitional cell carcinoma, is a malignant tumor characterized by local invasion and metastasis to the lymph nodes and distant sites.”

  1. 1, line 38-40.

3) Page 3, line 106 – “bladder wall invasion” – Can ultrasound truly be used to determine if invasion into the bladder wall is present?  Is it possible that some of the lesions classified as not muscle invasive were in fact muscle invasive but lacked a large enough invasion to be termed that via ultrasound exam alone?

[Response]                                                                                                                           

Thank you very much for your comments. As you mentioned, ultrasonography alone may underestimate the tumor's invasion into the muscle layer. Unfortunately, many of the cases included in this report did not undergo detailed pathological examination, so only those cases in which ultrasonography clearly showed invasion into the muscle layer were considered "invasive. In fact, we have added a limitation that more detailed pathological examination is required.

“Second, in this study, the presence or absence of tumor invasion in the muscle layer was evaluated only by ultrasonography, which may have missed invasion that is not actually visible on the images. Since pathological examination of all layers of the bladder is necessary to truly evaluate invasion into the muscular layer, it is necessary to in-crease the number of cases and evaluate it in combination with detailed pathological examination in the future.” P. 9, line 317-323.

4) Table 1. It appears that ALA intensity was low in 8/11 dogs treated with an NSAID vs. 2/7 dogs receiving no treatment vs 0/3 in dogs receiving prednisolone.  Were dogs assessed for differences between groups?  For male dogs, the ALA intensity was low in 8/14 dogs and for female dogs, ALA intensity was low in 3/7 dogs.  Were there significant differences based on sex?  

[Response]

Thank you for your comments. The intensity of 5-ALA fluorescence in the no-treatment group, the NSAIDs group, and the prednisolone group was again statistically compared, but no significant difference was found. Sex was also evaluated in the same way, but no significant difference was found. Added the following description and supplementary figures.

 “Sex and treatment status (no treatment, NSAIDs, prednisone) did not affect the fluorescence intensity of 5-ALA (supplementary figures 1 and 2).” P. 5, line 199-201.

5) Page 7, line 230: “we demonstrated that urothelial carcinoma cells in canine and human urine samples can be detected…”  Were there samples from humans tested?

[Response]

Thank you for your comments. The word "human" has been removed from this description because it was an error. P. 8, line 265.

6) Figure 2. This data should be broken up by treatments the dogs with urothelial carcinoma received (NSAID, prednisolone or no treatment) to determine if the treatment influenced the results.  There is a much wider range in the clinical dogs with urothelial carcinoma vs. the urothelial carcinoma cell lines, and it may be important to establish if there was any influence of the treatments these dogs already received.  This is particularly important given the known anti-tumor activity of NSAIDs in urothelial carcinoma and the widespread use of these medications in veterinary medicine once a dog is diagnosed with a suspected bladder tumor.

[Response]

Thank you for your comments. As mentioned earlier, we did not find that the treatment affected the fluorescence intensity of 5-ALA. This data has been added as a new supplementary figure.

7) Table 2. Again, I’d like clairification on how muscle-invasiveness was determined.  If no full thickness biopsy specimens were available, I’m concerned that US alone may underestimate muscle invasion.  It also may be important to evaluate dogs with prostatic involvement separately as this has historically been associated with a worse prognosis.

[Response]

Thank you for your comments. As mentioned above, the invasion into the muscle layer was only evaluated by ultrasonography in this study, so we added a description in the limitation section that it is necessary to conduct another study including pathological examination.

“Second, in this study, the presence or absence of tumor invasion in the muscle layer was evaluated only by ultrasonography, which may have missed invasion that is not actually visible on the images. Since pathological examination of all layers of the blad-der is necessary to truly evaluate invasion into the muscular layer, it is necessary to in-crease the number of cases and evaluate it in combination with detailed pathological examination in the future.” P. 9, line 317-323.

We have also added a new description of tumor location in Table 1. When we statistically evaluated the fluorescence intensity of 5-ALA in the vesical torigone group, urethra involvment group, and prostate involvment group based on the tumor location, there was no significant difference. We have added description and data as a new supplementary figure 3.

“In addition, there was no significant correlation between tumor location (vesical tori-gone, urethra involvement, prostate involvement) and fluorescence intensity by 5-ALA (Supplementary Figure 3).” P. 5, line 201-203.

8) With respect to the discussion and conclusions, I’m concerned that not enough variables were evaluated to determine if the high/low intensity is related to BRAF status, treatment (which many dogs had already had), tumor location (e.g., prostatic involvement), etc.  As with many studies looking to use urine for the purposes of diagnosing urothelial carcinoma, I am concerned about the lack of a group with inflammation (cystitis) and/or urinary tract infection present.  I would suggest evaluating for the same thing in groups of dogs with both of these conditions (or at least urinary tract infections).  Did any of the dogs with urothelial carcinoma have active urinary tract infections?  I think these questions need to be addressed before the discussion can truly be reviewed.  The authors acknowledge this as a limitation on page 8 beginning a line 283 and understand that benign conditions may mimic urothelial carcinoma.  It is essential that these sorts of conditions be included when developing this sort of diagnostic aid, otherwise potential false positives will be missed and the utility of the test will be overestimated.  At a minimum, the authors should perform additional analysis in urine samples from dogs with urinary tract infections and report if any of the dogs in this study had urinary tract infections.  Urinary tract infections are common and it should not be difficult to collect urine from multiple dogs with this condition in a short time frame.

[Response]

Thank you for your comments. As you pointed out, the authors themselves are well aware of the fact that the small number of cases is a limitation in this study, and this is described in the limitation section. Unfortunately, the hospital to which we belong is a secondary care facility and rarely receives cases of urinary tract infections that can be treated at a primary care facility, so we were unable to obtain samples of such cases for this study. In addition, the cases incorporated in this study did not show any findings that would raise suspicion of urinary tract infection. We have added a description about it.

“In these cases, there were no findings that would lead us to suspect a urinary tract in-fection.” P. 3, line 117-118.

Since this study is a pilot study, we plan to increase the number of cases and examine whether it is possible to differentiate urinary tract diseases other than tumors. We have added a description about it.

“Since this study is a pilot study, we plan to increase the number of cases to see if we can differentiate tumors from other urinary tract diseases.” P. 10, line 330-331.

Reviewer 2 Report

This manuscript explores the use of 5-ALA as a non-invasive method for diagnosis of canine urothelial carcinoma.  Overall the study population is small; however, this is simply a pilot study and respect that entirely.  The study in it's entirety has wonderful merit for a disease that can be frustrating to diagnose without more invasive procedures.  I have several line item revisions that should be addressed. 

Author Response

Reviewer: 2

1) Line 13-14: The use of the term “urinary cells from canine urothelial carcinomas” is confusing. Do you mean “from urine of patients with urothelial carcinoma”?  

[Response]

Thank you very much for your comments. As you pointed out, it means "from urine of patients with urothelial carcinoma". I rewrote it as follows.

“The addition of 5-ALA to tumor cells from urine of patients with urothelial carcinoma and canine urothelial carcinoma cell lines caused red fluorescence,” P. 1, line 13-14.

2) Line 26-27: Urinary cells sounds unusual. Throughout the paper, the term “urinary cells” is used. I would consider using “urine derived cells” instead.

[Response]

Thank you very much for your comments. I rewrote it as "Urine derived cells" as you suggested.

3) Line 39-41: I would disagree with this statement. Only about 16% of patients have evidence of metastasis at the time of diagnosis and it is not the presence of metastasis that prevents surgical intervention, it is the location with in the trigone of the bladder that typically removes surgery as a treatment option. Additionally, the reference for this statement (reference #2) is not an appropriate reference and should be removed. Reference #2 is pertaining to prostatic carcinoma which in fact does

have a much higher rate of metastasis in general. This statement needs to be reworded.

[Response]

Thank you very much for your comments. The relevant part has been rewritten as follows. The citation has also been replaced with an appropriate one.

“Diagnosis of canine urothelial carcinoma is often delayed due to nonspecific initial symptoms, and it has been reported that approximately 50% of cases have distant me-tastases at autopsy [2].” P. 1, line 40-42.

4) Line 43-44: Please rephrase with improved grammar.   

[Response]

Thank you for your helpful advice. The text of the relevant part was rewritten as follows.

“The most reliable test to diagnose urothelial carcinoma is histopathology, but it is a highly invasive test.” P. 2, line 47-48.

5) Line 45-50: These sentences should be reworded. Abnormal cells can also be detected cytologically in free catch urine. Line 45 should state that this is pertaining to cytological evaluation. I appreciated the comment about inflammation causing characteristics of malignancy, but would try to reword that

differently. I would remove “Therefore, cytological analyses of urinary cells often require the aid of skilled technicians” since I have worked with pretty skilled clinical pathologists who will not definitively diagnose UC on cytology.

[Response]

Thank you for your advice. As you pointed out, we have deleted and rewritten the relevant parts.

“The detection of abnormal epithelial cells in urine collected from patients by cytology can also aid in diagnosis. However, cytology alone cannot diagnose canine urothelial carcinoma because normal urothelial cells can also become morphologically abnormal like tumor cells due to prolonged contact with urine or inflammation” P. 2, line 58-60.

6) Line 72-73: Please reword to be clearer. Particularly “and blue-violet light is irradiated on the lesion area inducing red fluorescence in the tumor tissue”.  Consider “In human medicine, this property is widely used for photodynamicdiagnosis, in which 5-ALA is administered to patients with bladder cancer or gastric cancer, and red fluorescence is detected within the malignant lesion under the exposure of blueviolet light.

[Response]

Thank you for your advice. As you pointed out, we have deleted and rewritten the relevant parts.

“In human medicine, using this property of 5-ALA, photodynamic diagnosis is widely performed by administering 5-ALA to patients with bladder cancer or gastric cancer and irradiating them with blue-violet light to detect red fluorescence in the malignant lesion.”

  1. 3, line 80-85.

7) Line 74-77 I would combine these two sentences. Consider “ Furthermore, PpIX produces reactive oxygen species (ROS) when excited by red light at approximately 635nm and can be used as photodynamic therapy for the treatment of gliomas and gastric cancer”

[Response]

Thank you for your comment. The text of the relevant part was rewritten as follows.

“Furthermore, PpIX produces reactive oxygen species (ROS) when excited by red light at approximately 635nm and can be used as photodynamic therapy for the treatment of gliomas and gastric cancer” P. 2, line 87-89.

8) Line 83-85: Given this manuscript is regarding diagnostic use of 5-ALA, I would remove these lines pertaining to treatment with PDT.

[Response]

Thank you for your comment. I deleted the relevant part as you pointed out. P. 3, line 98-101.

9) Line 98: Why 21 dogs? Power analysis performed or just the number that presented during this time period? I suspect of course this is a pilot study for which I personally believe power analyses are pointless, but was just overall curious for this number?

[Response]

Thank you for your comment. As described in line100, the number of urothelial carcinoma cases that came to the hospital and had a urine sample available between 2020 and 2021 was simply 21. No selection of cases was made for inclusion.

10) Line 100-101: Please be more specific on location. Bladder, urethra, etc. Urothelial would also include the urothelial prostate.

[Response]

Thank you for your comment. Information on tumor location (vesical torigone, urethra involvement, prostate involvement) has been added to Table 1.

11) Line 107: Two periods after sentence

[Response]

I deleted it as you suggested.

12) Line 107-108: These are results and should be presented in the results section.

[Response]

Thank you for your comment. Removed and added the following text to the results section.

“We are also examining whether ALA fluorescence correlates with the patient's clinical condition, such as metastasis or invasion of the muscle layer. Seven of the 21 patients had metastasis in the lymph nodes and distant organs. Invasion of the bladder wall was observed in nine of the 21 cases.” P. 6, line 228-232.

13) Line 109-110: What are the demographics of the control group? Maybe consider adding them into Table 1. Did the control group have diagnostics performed to confirm their “healthy” status?

[Response]

Thank you for your comment. For healthy dogs, blood tests, urinalysis, and ultrasonography have been performed to confirm that they do not have any diseases. Information on healthy dogs has been added to Materials & Methods.

“The healthy dogs were all 8-year-old neutered males and were confirmed to be free of disease by blood tests, urinalysis and ultrasonography.” P.3, line 128-130.

14) Line 110-113: Was urine collected with the same method for the study population as it was the control population? This is not clear.

[Response]

Thank you for your comment. All cases and controls had their urine collected using a catheter under ultrasound guidance. The following statement has been revised.

“Urine samples were collected from cases and healthy dogs using a 4-6Fr catheter (Atom Medical Corporation, Tokyo, Japan) under ultrasound guidance.”

P.3, line 130-132.

15) Line 128: What are you using as a vehicle?

[Response]

Thank you for your comment. The vehicle uses ultra-pure water. The following description was added.

“or vehicle (ultra-pure water) for 2h…” P.4, line149.

16) Line 167: Did you do a nuclear stain (such as DAPI) to confirm this is all in the cytoplasm?

[Response]

Thank you for your comment. Unfortunately, we did not stain the nuclei in this study. There are several human studies in which nuclear staining has not been performed (Miyake et al., Cancer Sci. 2014. doi: 10.1111/cas.12393, Nakai et al., BMC Urol. 2014. doi: 10.1186/1471-2490-14-59.). Since red blood cells were removed from the sample, we believe that the stained cells are tumor cells or epithelial cells, and the cytoplasm is stained since the nucleus appears to be black.

17) Line 197: This should be reworded. 5-ALA fluorescence is not affecting clinical condition but rather is correlated to clinical status.

[Response]

Thank you for your comment. The following statement has been revised.

“We are also examining whether ALA fluorescence correlates with the patient's clinical condition, such as metastasis or invasion of the muscle layer.”

P.6, line 228-229.

18) Line 230-231: Please remove the part about human urine. This study did not directly evaluate human urine samples.  

[Response]

Thank you for your comment. This statement was deleted because it was incorrect.

P.8, line 265.

19) Line 237: Please change “cat and dog” to “feline and canine”

[Response]

Thank you for your comment. I fixed it as you suggested.

P.8, line 272.

20) Line 282: It is important to know how the tumor tissues were obtained, whether by cystoscopy or cystotomy. If obtained by cystoscopy, how can we be certain that the “non-muscle” invasive cases were not just false negative due to superficial sampling of just the mucosal layer? The overall number of nonmuscle-invasive cases in this study seems to be quite low given that the majority of canine cases are muscle invasive. This certainly needs to be elaborated on in the methods section as well as the limitations if these were collected by cystoscopy.

[Response]

Unfortunately, we were not able to perform a detailed pathological search in most of the cases, so we consider an invasion into the muscle layer to be an "invasion" when it is clearly visible on ultrasonography. As the reviewer mentioned, pathological examination of all layers of the bladder is necessary to truly evaluate invasion of the muscle layer, and this study may have underestimated invasion of the muscle layer. The authors believe that the results are worthy of publication, but would like to leave the final judgment to the editor. In addition, the following text has been added to the section on limitation.

“Second, in this study, the presence or absence of tumor invasion in the muscle layer was evaluated only by ultrasonography, which may have missed invasion that is not actually visible on the images. Since pathological examination of all layers of the bladder is necessary to truly evaluate invasion into the muscular layer, it is necessary to in-crease the number of cases and evaluate it in combination with detailed pathological examination in the future.” P. 9, line 317-323.

Reviewer 3 Report

Recommendation: Minor revision

  In this manuscript entitled “Detection of canine urothelial carcinoma cells in urine using 5-aminolevulinic acid”, the authors propose that 5-aminolevulinic acid (5-ALA) is a novel non-invasive diagnostic tool for canine urothelial carcinoma. The primary finding is that urinary tumor cells can be detected with high sensitivity and specificity by 5-ALA fluorescence intensity, which has not been reported in veterinary medicine. The reviewer also thinks a new tool for diagnosis of urothelial tumor may be of interest to the veterinary field because histopathological examination of tumor tissues is invasive. Overall, this is a well-written manuscript and the data support the main conclusion. I have a few minor comments that the authors might consider, explained below.

  1. Line 85: “..” => “.”
  2. On method section “cell culture with 5-ALA”, the authors have set the incubation time of 5-ALA to 2 hours. Is this incubation time optimal? Is there proper reference? If the authors have already performed some experiments to determine the culture condition, the details should be addressed on method section.
  3. Line 112: Is “1500xg” correct as a rotation speed? Isn’t it “rpm”?
  4. On Discussion section, the authors mentioned there are no reports on the direct relationship between BRAF mutation and 5-ALA. I also think this is the next issue to be addressed. In this study, the authors demonstrated that the intensity of 5-ALA fluorescence was positively correlated with the increased tumor malignancy. In addition, the high fluorescence intensity of 5-ALA was observed in BRAF mutation cases. Is there a relationship between BRAF mutation and tumor malignancy in canine urothelial carcinoma? If so, it may suggest a relationship between BRAF mutation and 5-ALA.

Author Response

Reviewer: 3

1) Line 85: “..” => “.”  

[Response]

Thank you for your comment. This statement was deleted because it was incorrect.

2) On method section “cell culture with 5-ALA”, the authors have set the incubation time of 5-ALA to 2 hours. Is this incubation time optimal? Is there proper reference? If the authors have already performed some experiments to determine the culture condition, the details should be addressed on method section.  

[Response]

Thank you for your comment. Since the study of 5-ALA in human bladder cancer was conducted with an incubation time of 2 hours, we decided to use a similar incubation time in this study. The following text was added along with additional citations.

“Based on previous reports[15], cells were cultured in RPMI medium without FBS and 1 mM 5-ALA (Neopharma Japan, Tokyo), or vehicle (ultra-pure water) for 2 h at 37°C and 5% CO2.” P.5, line 147.

3) Line 112: Is “1500xg” correct as a rotation speed? Isn’t it “rpm”?

[Response]

Thank you for your comment. As you pointed out, 1500 rpm is correct. I have corrected it.

4) On Discussion section, the authors mentioned there are no reports on the direct relationship between BRAF mutation and 5-ALA. I also think this is the next issue to be addressed. In this study, the authors demonstrated that the intensity of 5-ALA fluorescence was positively correlated with the increased tumor malignancy. In addition, the high fluorescence intensity of 5-ALA was observed in BRAF mutation cases. Is there a relationship between BRAF mutation and tumor malignancy in canine urothelial carcinoma? If so, it may suggest a relationship between BRAF mutation and 5-ALA.

[Response]

Thank you for your comment. Last year, a report was published on a large retrospective study of the correlation between BRAF mutations and prognosis in canine urothelial carcinoma, which stated that BRAF mutations do not correlate with prognosis (Gedon et al., Vet Comp Oncol. 2021. doi: 10.1111/vco.12790). However, our results suggest that BRAF mutations may be associated with pathways involved in 5-ALA metabolism, and these abnormalities may affect the pathogenesis of tumors, which will be investigated in more detail in the future.

Round 2

Reviewer 1 Report

Thank you for addressing most of my concerns from the first review.  I strongly believe that any follow up study must include samples from dogs with known cystitis and/or urinary tract infection.

There are some minor English-language/grammatical issues that should be addressed during copy-editing.

Page 9, line 296 - There are two commas after the word "Third".

Page 9, line 303 - There seems to be an extra phrase inserted, "Diagnosis of canine urothelial carcinoma is often delayed" at the start of a different sentence.

Author Response

Manuscript ID: animals-1560100

Title: Detection of canine urothelial carcinoma cells in urine using 5-aminolevulinic acid

Kenjiro Kaji, Tomohiro Yonezawa, Momoi Yasuyuki, Shingo Maeda

Dear editor,

We are resubmitting this manuscript to Animals after carefully considering the suggestions made by reviewers and editors. Revisions we have made are as follows:

Reviewer: 1

1) Thank you for addressing most of my concerns from the first review.  I strongly believe that any follow up study must include samples from dogs with known cystitis and/or urinary tract infection.   

[Response]

Thank you very much for your comments. As you mentioned, we will proceed with the search as soon as we have samples of cystitis cases and urinary tract infection cases.

2) Page 9, line 296 - There are two commas after the word "Third".

[Response]

Thank you very much for your comments. Removed commas in the relevant sections.

3) Page 9, line 303 - There seems to be an extra phrase inserted, "Diagnosis of canine urothelial carcinoma is often delayed" at the start of a different sentence.

[Response]                                                                                              

Thank you very much for your comments. I deleted the relevant sentence and replaced it with the following sentence.

“Since this study is a pilot study, we plan to increase the number of cases to see if we can differentiate tumors from other urinary tract diseases.” P. 12, line 444-445.

To show which parts have been changed, the added and removed parts (marked in yellow) are shown. With these changes, we have addressed all the suggestions made by the reviewers. We would like to thank the reviewers for their helpful suggestions. We hope that this manuscript will be published in Animals.

Sincerely,

Kenjiro Kaji

Shingo Maeda